# Poly-ε-Caprolactone-Hydroxyapatite-Alumina (PCL-HA-α-Al_2_O_3_) Electrospun Nanofibers in Wistar Rats

**DOI:** 10.3390/polym14112130

**Published:** 2022-05-24

**Authors:** Luis Roberto Ruiz-Ramírez, Oskar Álvarez-Ortega, Alejandro Donohue-Cornejo, León Francisco Espinosa-Cristóbal, José Rurik Farias-Mancilla, Carlos A. Martínez-Pérez, Simón Yobanny Reyes-López

**Affiliations:** 1Departamento de Ciencias Químico Biológicas, Instituto de Ciencias Biomédicas, Universidad Autónoma de Ciudad Juárez, Ciudad Juárez 32315, Mexico; al199127@alumnos.uacj.mx (L.R.R.-R.); al204562@alumnos.uacj.mx (O.Á.-O.); 2Departamento de Estomatología, Instituto de Ciencias Biomédicas, Universidad Autónoma de Ciudad Juárez, Ciudad Juárez 32315, Mexico; adonohue@uacj.mx (A.D.-C.); leon.espinosa@uacj.mx (L.F.E.-C.); 3Institute of Engineering and Technology, Autonomous University of the City of Juárez, UACJ, Ciudad Juárez 32310, Mexico; rurik.farias@uacj.mx (J.R.F.-M.); camartin@uacj.mx (C.A.M.-P.)

**Keywords:** biocompatibility, poly-ε-caprolactone, hydroxyapatite, alumina, foreign body reaction

## Abstract

Biodegradable polymers of natural origin are ideal for the development of processes in tissue engineering due to their immunogenic potential and ability to interact with living tissues. However, some synthetic polymers have been developed in recent years for use in tissue engineering, such as Poly-ε-caprolactone. The nanotechnology and the electrospinning process are perceived to produce biomaterials in the form of nanofibers with diverse unique properties. Biocompatibility tests of poly-ε-caprolactone nanofibers embedded with hydroxyapatite and alumina nanoparticles manufactured by means of the electrospinning technique were carried out in Wistar rats to be used as oral dressings. Hydroxyapatite as a material is used because of its great compatibility, bioactivity, and osteoconductive properties. The PCL, PCL-HA, PCL-α-Al_2_O_3_, and PCL-HA-α-Al_2_O_3_ nanofibers obtained in the process were characterized by infrared spectroscopy and scanning electron microscopy. The nanofibers had an average diameter of (840 ± 230) nm. The nanofiber implants were placed and tested at 2, 4, and 6 weeks in the subcutaneous tissue of the rats to give a chronic inflammatory infiltrate, characteristic foreign body reaction, which decreased slightly at 6 weeks with the addition of hydroxyapatite and alumina ceramic particles. The biocompatibility test showed a foreign body reaction that produces a layer of collagen and fibroblasts. Tissue loss and necrosis were not observed due to the coating of the material, but a slight decrease in the inflammatory infiltrate occurred in the last evaluation period, which is indicative of the beginning of the acceptance of the tested materials by the organism.

## 1. Introduction

Human beings have the need to repair the loss of tissues caused by trauma, diseases, or deterioration. One of the ways to correct the loss of tissue is through autologous implants. However, these types of implants have limitations in relation to their use, such as the amount of material available, the surgical procedures required to obtain the implant, complications in healing, and even infection in the donor wound sites [1,2]. Therefore, in recent years, research has focused on the development of advanced materials that have characteristic properties such that they can reproduce the function of living tissues in a safe and physiologically acceptable way. The range of materials that can reproduce the function of a living tissue in a safe way are known as biomaterials [3]. The main characteristic of a biomaterial is that it should not cause major physiological alterations when used in a biological system. A biomaterial must meet certain requirements such as short- and long-term biostability—the host must not react aggressively to the presence of the material and must maintain its physical and chemical properties for as long as the biomaterial remains in the biological system; these requirements define the concept of biocompatibility [4,5].

For a material to be used in a human being, it must be biocompatible. In determining biocompatibility, different tests are required that can define whether it is suitable for use in a biological system. Among the tests used to determine the biocompatibility of a material are mechanical simulations, in vitro tests, toxicological tests, and tests on experimental animals. The most used preliminary biocompatibility tests are the MTT and proliferation tests in fibroblasts, which, in turn, provide an overview of the possible effect that the material could have on an experimental animal, which is the subsequent step to in vitro tests [5,6]. In tests on experimental animals, the material is placed in the subcutaneous tissue for a defined period while observing the reactions that the material provokes on the animal and the tissues adjacent to the implantation area. Together, these tests provide valuable information on the effects that the material can cause under different conditions; therefore, they are a fundamental part of defining the biocompatibility of a material. Considering the complexity of tissue reactions and the great diversity of existing materials, it is said that the ideal biomaterial must have a dynamic surface such that it does not generate changes at the histological level, for example, collagen sutures that are reabsorbed after the completion of their function, without causing histological changes; this is why materials that are similar to those present in the human body are sought, generating greater biocompatibility [7].

Hydroxyapatite is a material with biomedical applications due to its great compatibility, bioactivity, osteoconductive properties, and great similarity to the inorganic phase of human bone [8]. One of the advantages of having a hydroxyapatite-based material is that it tends to become part of the bone due to its great porosity, which is why it is used as an orthopedic material [9] and in dental applications for the replacement of autologous and allogeneic bone grafts. Among other applications is its use as a bone cement in the repair of craniofacial defects and graft material for the augmentation of the maxillary sinus floor and the proximal tibia [10]. However, despite its great biocompatibility, hydroxyapatite has great fragility; therefore, its mechanical stability limits its use for the regeneration of bone defects [11]. The fragility of hydroxyapatite can be reduced with the addition of ceramics such as alumina, which is known for its high hardness and chemical stability, which can enhance the properties of hydroxyapatite [9]. The properties of alpha alumina are mainly determined by the crystal structure, with alpha phase being the most stable. Within the main characteristics of alpha alumina (α-Al_2_O_3_) is its high hardness; resistance to abrasion; and high chemical, electrical, and mechanical resistance. Currently, alpha alumina has been recognized as a biocompatible ceramic for biomedical applications, as well as the development of both dental and prosthetic implants—it has even been used as a substitute for metallic materials thanks to its hardness [12]. Finally, with the presence of biocompatible particles that help in osteointegration, there is the need for a matrix if they are in the form of nanoparticles and not as solid scaffolds, hence the need for the use of polymers such as poly-ε-caprolactone (PCL). Poly-ε-caprolactone is one of the most widely used biodegradable synthetic polymers in tissue engineering; its chemical properties favor hydrolytic degradation and once degraded, its monomeric components are eliminated by natural means. Although, PCL can cause an inflammatory response due to its acid breakdown and can take several years to degrade compared with other biodegradable synthetic polymers, with its degradation being a function of its molecular weight. PCL is considered as a drug delivery system, it is also used to improve bone growth and regeneration in the treatment of bone defects [13,14,15]; thus, the addition of ceramic particles using this polymer as a matrix can favor the PCL degradation and improve the treatment of bone defects. Electrospinning is an operative method to produce polymer scaffolds, which permits the production of fibers at the nano and microscale from natural and synthetic polymers, increasing the possibility of mechanical and biological properties desired [14,15]. The production of bioscaffolds is sought after for many biomedical applications, including bones, musculoskeletal tissues, nerves, and blood vessels. The proposed materials and their processes must meet and have special characteristics such as the presence of fibers oriented and organized according to which is desired to implant, and appropriate mechanical properties [16]. Studies examining compounds such as hydroxyapatite ceramic particles and biopolymers have revolutionized bioresorbable bone tissue engineering. The mechanical behavior of electrospun PCL fibers has already been extensively studied; hence, providing the starting point to study the effect of the incorporation of osteoconductive particles such as hydroxyapatite. Studies in electrospun PCL/TCP compounds show a decrease in their mechanical properties, which is why the incorporation of particles that increase this resistance is sought without affecting the biocompatible properties of the scaffold, as is the case of alpha alumina. Alpha alumina has a chemical stability under adverse conditions such as strong acidic or alkaline media. In search of scaffold with the desired biocompatible properties, we have worked on the synthesis and characterization of PCL, HA, and alpha alumina fibers, reporting the reaction conditions and obtaining parameters of the fibers, followed by the study of cell viability tests in the membranes, finding that scaffolds obtained are biocompatible and promote cell proliferation [17]. The principal aim of this project is to show the preliminary Biocompatibility Tests for Poly-ε-Caprolactone, Poly-ε-Caprolactone-α-alumina, Poly-ε-Caprolactone-Hydroxyapatite, and Poly-ε-Caprolactone-α-alumina-Hydroxyapatite fibers in Wistar Rats to a develop a biocompatible composite.

## 2. Materials and Methods

### 2.1. Synthesis of Hydroxyapatite Nanopowders

Hydroxyapatite powders were prepared by the chemical precipitation method. Ca(NO_3_)_2_·4H_2_O (ACS, Sigma-Aldrich © ≥99%, St. Louis, MO, USA) and (NH_4_)_2_HPO_4_ (ACS, Alfa Aesar © ≥99%, Haverhill, MA, USA) were dissolved in deionized water separately; the phosphate solution was then added dropwise to the nitrate solution and mixed with a magnetic stirrer for 60 min. The pH of the final solution was adjusted to 11 using NH_4_OH (Jalmek^®^ ≥99%) and stirred for 60 min at room temperature. The precipitate formed was separated from the liquid by centrifugation at 12,500 rpm and washed twice with deionized water and once with isopropanol, centrifuging between each washing. The precipitate was heat-treated in a high temperature oven (Thermo Scientific © Heratherm™, Waltham, MA, USA) at 100 °C for 24 h and then calcinated at 800 °C for two hours in an oven (Thermo Scientific © Thermoline™) using a ramp of 5 °C/min. After calcination, the fragile agglomerates were grinded in an agate mortar to obtain fine powders according to a previously reported method [17].

### 2.2. Synthesis of Alpha Alumina Nanopowders

The synthesis of alpha alumina nanopowders was carried out from the preparation of emulsions with a organometallic precursor of aluminum formate and urea, using absolute ethanol (≥96%) as solvent, until a homogeneous white paste was obtained. The aluminum formate (Al(O_2_CH)_3_) was obtained using the chemical synthesis described by Reyes-López et al. [12] that involves a mixture of aluminum and formic acid (HCOOH) using mercury chloride (HgCl_2_) as a catalyst to obtain the aluminum formate solution that was dried by spraying to produce a fine granulated organometallic precursor. The heat treatment was carried out by microwave combustion. Each of the emulsions was exposed to microwaves (1000 watt) for 5 min to obtain porous agglomerates. The agglomerates were calcined in a muffle furnace at a temperature of 1050 °C for one hour in an atmosphere rich in oxygen, using a ramp of 10 °C/min. After calcination, the brittle agglomerates were grinded in an agate mortar to obtain fine alpha alumina powders.

### 2.3. Preparation of Solutions to Obtain the Nanofibers

The solutions were prepared from the dilution of poly-ε-caprolactone (PCL) (ACS, Sigma-Aldrich ©, 80,000 kDa) in 10% in acetone; then, 2% HA powder and 2% α-Al_2_O_3_ powder were added, and a solution of PCL with HA and PCL with α-Al_2_O_3_ were also prepared. The solutions were kept under magnetic stirring for one hour and in an ultrasound bath for 1 h. The precursor solutions were transferred to 10 mL glass syringe (KD © Scientific™, Holliston, MA, USA). In the electrospinning process, the distance between the needle and the collector was 10 to 14 cm, using a feed flow of 10 to 16 μL/min, and the voltage used was 8 to 11 kV. The environmental parameters during the electrospinning process were 21 °C and 35–42% for temperature and relative humidity, respectively; this was described by Reyes-López et al. for the in vitro evaluation of poly-ε-caprolactone-hydroxyapatite-α-alumina electrospun fibers on the fibroblast’s proliferation [17]. Four different composites were obtained by means of the electrospinning technique: PCL, PCL-HA, PCL-α-Al_2_O_3_, and PCL-HA-α-Al_2_O_3_. The PCL, PCL-HA, PCL-α-Al_2_O_3_, and PCL-HA-α-Al_2_O_3_ fibers obtained by the electrospinning technique were characterized by infrared spectroscopy (FTIR) using an Alpha platinum-ATR Bruker spectrometer (Bruker Corporation, Billerica, MA, USA). For scanning electron microscopy (SEM), a JEOL JSM-6400 device (JEOL Ltd., Tokyo, Japan) operating at 20 kV and equipped with an X-ray scattering spectrometer (EDX) was used. The magnifications used were 5000×; 10,000×; and 20,000×. For each characterization technique, a sample of each fiber measuring 1 cm × 1 cm was used. The exhaustive characterization of particles and fibers was already reported in the first study of these fibers; so, in this work, only a short characterization of scaffolds is presented [17].

### 2.4. In Vivo Biocompatibility Test

Wistar rats (four- or three-month-old males), weighing approximately from 150 to 300 g, were used, divided into three groups, and one rat was used as a control. Each of the rats was housed individually in conditions established by the American Veterinary Medical Association’s (AVMA), NOM-062-ZOO-1999, and the Institutional Animal Care and Use Committee (IACUC), Comite Institucional de Etica y Bioetica, Universidad Autonoma de Ciudad Juarez (CIEB-UACJ). The technical specifications for the production, care, and use of laboratory animals were adhered to during the entire process of the experimental phase (date of committee approval—23 October 2017, project—CIBE-2017-2-84). It needs to be stated that the rats were fully anesthetized and the personnel who performed the procedure were fully trained. Three fractions of each of the nanofibers were taken. The implants measured approximately 8–10 mm in length by 1.3 mm in internal diameter. Once the implants were made, they were UV-sterilized for half an hour to avoid any type of contamination during the implantation process. Figure 1 shows the morphology of the implant before being implanted in the animal.

### 2.5. Implant Placement

For the anesthetic process, an intramuscular injection of ketamine/xylazine (0.12/0.01 mL, PiSa Agropecuaria, Hidalgo, Mexico) was performed in one of the hind legs of the rats. After anesthetizing the rats, the dorsal area was shaved and then covered with iodine (Dynarex, Orangeburg, NY, USA, ≥99%) to avoid any infection during the surgery. After applying anesthesia, the material was implanted. With the help of a previously sterilized veterinary surgical kit, 4 incisions were made in the dorsal part of the rats. Once the materials were implanted, the wounds were sutured with 2 points per incision.

### 2.6. Histological Samples

Euthanasia was performed at 2, 4, and 6 weeks after implantation (one rat every 2 weeks). Intravenous injection of sodium pentobarbital is the preferred method for euthanizing horses, dogs, cats, and rodents, causing a quick and painless death of the body [16]. An overdose of sodium pentobarbital (PiSa Agropecuaria) was administered intraperitoneally to rats. Small incisions were made in the implantation areas. Four biopsies were taken, one from each material obtained, and deposited in formalin (Sigma-Aldrich, ≥99%) for preservation. The criteria for euthanasia coincided with the experimental endpoints to reduce the pain or distress caused by the experimental condition. Once the animals’ tissues were obtained, they were placed in 99% formalin (Sigma-Aldrich) to preserve and fix the tissue until its subsequent staining. The pathological changes observed in the animal for the effects of the treatment were carried out prior to the main experiment to allow better results and to minimize the animals’ pain while the study was completed. All veterinarians performing euthanasia needed to be properly trained.

For inclusion in paraffin, the biopsies were placed in cassettes to be subsequently dehydrated by ethanol (pure solution, 96%) with a gradation increasing to 100%. After doing so, the samples were transferred to xylene (Sigma-Aldrich, ≥99%), which was used as an intermediate reagent. The cassettes were taken and transferred to paraffin previously liquefied in an oven. Liquid paraffin was poured into various molds. Samples were introduced and refrigerated at 4 °C for hardening. The paraffin blocks with the tissues were cut to a thickness of 5 µm and, once the cuts were obtained, they were placed in a 37 °C water bath to warm them up. Afterwards, the cuts were placed in an oven at 60 °C for the deparaffinization process. Tissues were rehydrated with decreasing ethanol until they reached water. Tissues were dipped in hematoxylin (Sigma Aldrich) for 3 min, washed with water, then dipped in eosin red (Sigma Aldrich) for 30 s and washed with ethanol to remove traces of dyes. Tissues were dehydrated again with increasing ethanol. Once the coverslip was placed on the slide with the histopathological cut, the tissues were observed under a microscope. Finally, the plates were observed in an Oxion light microscope at 10× and 40× and photos were taken by a Cemex camera (Euromex, IL, USA).

## 3. Results and Discussion

The synthesized α-Al_2_O_3_ powders were characterized by SEM, EDX, and dynamic light scattering (DLS). In Figure 2a,b, the powders obtained from α-Al_2_O_3_ are appreciated and the existence of nanoparticles between 35 and 62 nanometers in diameter is observed. This can be corroborated with the dynamic light scattering analysis (Figure 2c), where the results indicate that most of the nanoparticles of α-Al_2_O_3_ powders have an approximate size of 30 to 100 nanometers. The chemical composition of the nanopowders of α-Al_2_O_3_ was determined by means of an EDX (Figure 2d) analysis, in which it is confirmed that the elements present in the synthesized nanopowders are aluminum and oxygen. The characterization of hydroxyapatite nanopowders by SEM shown in Figure 3a,b reveal that the synthesized nanoparticles are spherically shaped and are of nanometer to submicron size. Dynamic light scattering analysis in Figure 3c shows that most of the nanoparticles have a size of 100 to 200 nanometers. The chemical composition of the hydroxyapatite nanopowders by EDX (Figure 3d) confirms that the elements present in the synthesized nanopowders are calcium, phosphorus, and oxygen, and are constituent elements present for hydroxyapatite structure.

In Figure 4a, the α-Al_2_O_3_ phase is fully identified by X-ray diffraction patterns; the diffractogram shows peaks located at 35°, 37.5°, 43°, 52°, 57°, 61.5°, 66°, and 68° that can be associated with the (104), (110), (113), (024), (116), (018), (214), and (300) planes, respectively (JCPDS 10-0173) [12,17]—and for hydroxyapatite, the diffraction pattern discloses peaks in 2θ at 18.3°, 23.1°, 26°, 29.2°, 32.2°, 34.3°, 40.1°, 47°, 49.5°, 54.4°, and 64.2° due to the presence of the crystallographic planes (101), (111), (200), (120), (211), (202), (310), (222), (213), (400), and (304) are related to the hexagonal structure of hydroxyapatite according to the diffraction pattern (R050512) [15,17,18]. In Figure 4a, for the infrared spectrum of hydroxyapatite in black, different absorption bands of the different functional groups present in hydroxyapatite can be observed at 550, 602, 960, and 1020 cm^−1^ for the PO_4_^−3^ group. The bands at 723 and 1020 cm^−1^ are bands belonging to the pyrophosphate groups (P_2_O_7_^−4^) and hydrogen phosphate (HPO_4_^−2^), respectively [16]. Figure 4b (red spectra) shows the infrared spectrum for the α-Al_2_O_3_ nanopowders for bands corresponding to the flexural stretching of the Al–O bond at 440 cm^−1^ and bands corresponding to the tension stretching of the Al group. For Al–O bond, bands at 440, 570, and 640 cm^−1^ are prominent characteristics for the alpha phase of alpha alumina [19]. The obtained nanofibers from the electrospinning technique show white color membranes in rough form due to the presence of ceramic nanoparticles. The infrared spectrum of Figure 4c shows that the infrared bands of all the membranes of the nanofibers obtained have the same wavelengths corresponding mainly to the characteristic spectrum of PCL. This is due to the fact that the ceramic nanoparticles of α-Al_2_O_3_ and hydroxyapatite are lesser in proportion in the polymer matrix. An additional suggestion is that the ceramic particles remain embedded in the polymer, which does not allow the interaction of infrared radiation; therefore, it is not possible to observe the characteristic vibrations of ceramic particles such as α-Al_2_O_3_ and hydroxyapatite, as reported previously [12,17]. The vibrational bands, characteristic of the PCL, give asymmetric stretching vibrations at 2943 cm^−1^ for the CH_3_ group and deformation stretching at 2865 cm^−1^ for the methylene group; vibrational stretching bands at 1720 cm^−1^ for C=O, bands stretching at 1365 and 1470 cm^−1^ for the COC group, and vibration bands by asymmetric stretching at 1240, 1110, 1165, 960, 732, and 450 cm^−1^ for the COC group; and vibrations at 2800 and 2650 cm^−1^ bands corresponding to the -CH_2_ group [18,19,20,21].

Figure 5 shows the SEM micrographs of each of the manufactured scaffolds; Figure 5a shows 10% PCL nanofibers, Figure 5b shows 10% PCL nanofibers with 2% HA, Figure 5c shows 10% PCL nanofibers with 2% α-Al_2_O_3_, and Figure 5d shows 10% PCL nanofibers with 2% HA and 2% α-Al_2_O_3_. The images obtained by SEM show that the nanofibers containing PCL and ceramic particles with different diameters are unidirectional. Fibers with PCL polymer (Figure 5a) show a morphology with random arrangement; a diameter of (0.84 ± 0.23) microns; and a smooth, cylindrical surface free of beads, precipitates, and fractures. The PCL-HA nanofibers are shown in Figure 5b, where it is observed that the hydroxyapatite is embedded in the PCL reaffirming the presence of the dispersed hydroxyapatite in the polymeric solution. It was observed that the nanofibers presented with a smooth morphology and a size of approximately (0.96 ± 0.32) μm. The PCL-α-Al_2_O_3_ nanofibers are shown in Figure 5c; in the nanofibers, defects can be observed that correspond to the α-Al_2_O_3_ particles embedded in the PCL; this causes an increase in the diameter of the nanofibers. The morphology of the nanofibers can be observed as rough, continuous, randomly distributed, and with variable diameter. The average diameter of the nanofibers is (1.13 ± 0.41) μm; in addition, some agglomerations of ceramic material can be observed. Figure 5d shows the micrograph of the PCL-HA-α-Al_2_O_3_ scaffold. A contrast is observed between the HA and α-Al_2_O_3_ particles, where the HA particles present a characteristic brightness in contrast to the α-Al_2_O_3_ particles that are observed as opaque; the fibers present a random distribution with a rough surface due to the encrustation of the ceramic particles and have an approximate diameter of (1.39 ± 0.64) μm.

Preliminary in vivo biocompatibility tests in subcutaneous tissue of Wistar rats for the four different materials (PCL, PCL-HA, PCL-HA-α-Al_2_O_3_, and PCL-HA-α-Al_2_O_3_) for six weeks show differences when performing the evaluation of the inflammatory infiltrate in the contact area of the material every two weeks. The initial placement of the implant of the material through an incision causes injury to the animal with a natural inflammatory response by the body, due to loss in the continuity of the tissue because of the injuries caused in the vasculature found in the subcutaneous connective tissues [22]. The inflammation process begins as an early response to the presence of a foreign body; the inflammation process includes the production of proinflammatory cytokines such as IL-1, IL-2, and IL-6, which in turn, participate in the activation of the immune system and in the release of prostaglandins and chemotactic substances [23].

Image 6 shows a microscopic image of the subcutaneous tissue of Wistar rats when exposed to PCL. At day and week zero in Figure 6a, there is no form of inflammatory infiltrate and adipose tissue, and muscle tissues are appreciated. The micrograph for 2 weeks of exposure to PCL is shown in Figure 6b and for 4 weeks of exposure to PCL in Figure 6c,d, corresponding to 6 weeks of exposure to PCL. It is observed that at two weeks, there is a chronic inflammatory infiltrate with the main presence of lymphocytes, multinucleated giant cells—cells that are formed at the moment in which a macrophage tries to engulf a body that is larger than it; in order to do so, it fuses with other macrophages, thus generating the multinucleated giant foreign body cells characteristic of the presence of a foreign material in the area of inflammation. This type of inflammatory infiltrate is maintained throughout the investigation period; however, no damage or loss of tissue is observed, which is why it is considered that the material is not toxic since it presents an inflammatory process similar to that of a common lesion. The inflammatory infiltrate does not diminish over the weeks since the presence of the implant in the tissue causes the inflammatory and repair process to be prolonged, producing a fibrous layer around the implant [24,25].

Another reason why the inflammatory infiltrate does not diminish over the weeks is due to the shape of the implant used in the form of a compact tube, which avoids contact of the material with living tissue and its fluids; therefore, the cells present in the medium identify it as a foreign body unleashing a chain of signals from the immune system, reaching the implant area sending cells to try to eliminate it; a greater integration of the material would have been achieved by putting the membrane without rolling, taking advantage of the area of contact and flow of biological substances through nanofiber networks. The use of biocompatible ceramic particles favors the biocompatibility of nanofibers, for example, hydroxyapatite increases cell conduction and induction capacity generates greater biocompatibility; as they are composed of Ca^2+^ ions, they help in the suspension of HPO_4_^2−^, allowing the assimilation of ions. The establishment of these chemical exchanges allows the formation of interfacial bonds in living tissue, favoring the process of integration and tissue formation [21,26].

Figure 7 shows the tissues exposed to the PCL-HA material where, throughout the experimentation period, the inflammation remained in a chronic inflammatory infiltrate, as in Figure 6. However, by taking advantage of the properties of hydroxyapatite inflammation at 6 weeks (Figure 7d), we can see that there is a slight decrease in the inflammatory infiltrate; the same effect was observed in tissues exposed to PCL-α-Al_2_O_3_ (Figure 8). Figure 9 shows the tissues exposed to the PCL-HA-α-Al_2_O_3_ composite where the inflammatory effect is chronic throughout the investigation period. However, as in the case of the PCL-HA and PCL-α-Al_2_O_3_ in the last period of 6 weeks (Figure 9d), a slight decrease in the inflammatory infiltrate is observed, which is attributed to the fact that this composite has the characteristics of hydroxyapatite, α-Al_2_O_3_, and PCL; therefore, it would be more likely to generate biocompatibility as mentioned in some investigations where in vitro biocompatibility tests have been conducted [17]. In previous results from the cell viability test carried out on the four membranes, they were found to be biocompatible and to favor cell proliferation. For the PCL membrane, the viability values are 14 ± 1.7, 22 ± 2.4, and 127 ± 5.1%, more than those of the control cells, at 24, 48, and 72 h of incubation, respectively. The PCL-HA presented increments of 14 ± 3.1, 89 ± 4.8, and 131 ± 4.4% at the same times. The PCL-α-Al_2_O_3_ particles presented increments of 43 ± 2.3, 39 ± 3.6, and 119 ± 5.8% on cell growth at the three times of incubation with respect to the control; finally, the viability on the PCL-HA-α-Al_2_O_3_ reached values of 52 ± 3.6, 61 ± 3.1, and 138 ± 4.4%, higher than those of the control at 24, 48, and 72 h of incubation (Table 1) [17].

The immunological reactions observed in the tissues exposed to the evaluated materials are characteristic of a foreign body reaction, which indicates a probable rejection of the evaluated materials by the Wistar rats. As previously explained, one of the reasons causing the foreign body reaction could be the excessive amount and form of implanted material, as observed in this investigation. Once a lesion is generated, protein adsorption is essential to promote good healing and trigger the different pathways involved in the healing process; the most important pathways are the coagulation pathway by the complement system. From the fibrinolytic system and platelet aggregation [26], once the pathways are activated, the generation of a provisional matrix begins at the place of implantation of the material, mainly fibrin, which is responsible for initiating the angiogenesis process to improve tissue healing [23]. Afterwards, there is an acute inflammation of short duration, where the migration of leukocytes, mainly neutrophils, occur [15,17,22,24,27,28,29]. As the implanted material is of excessive dimensions, it truncates the adsorption of proteins and, with this, the healing pathways, thereby generating foreign body reactions that could have been avoided if the implanted material had been dosed in a good way and applied in the form of a membrane.

Once neutrophils migrate to the implant area, they try to engulf the material, an action that is not possible since the material is larger than the neutrophil, this process is known as “frustrated phagocytosis”. Failure to achieve phagocytosis begins a type of chronic inflammation such as the one that can be seen in Figure 6, Figure 7, Figure 8 and Figure 9 with the presence of lymphocytes, monocytes, macrophages, plasma cells, and multinucleated giant cells; when these cells reach the area of the implant, a fibrous layer is generated with the intention of isolating the material from the organism [30,31]. Although this preliminary study cannot be used to determine if the materials (PCL, PCL-HA, PCL- α-Al_2_O_3_, and PCL-HA- α-Al_2_O_3_) are fully biocompatible with the subcutaneous tissue of Wistar rats, it can be said that at least there is no damage observed in the tissues adjacent to the implant or loss of tissue, in addition to the fact that, as previously explained, the incorporation of ceramic particles to the PCL helps to slightly decrease the degree of inflammatory infiltrate at 6 weeks, which could be an indication of acceptance by the body to the material.

Comparing the infrared spectrum (Figure 10) and SEM micrograph (Figure 11) of fiber composites analyzed, the addition of cells resulted in the formation of amide bands for collagen; the intensity of the amide band increased, indicating a certain degree of interaction between the composites and collagen according to index (IP = 1665/1720) for the formation of the collagen with time. According to previous results on PCL, at 48 h after the seeding of the cells, a mineralization process began with the formation of calcification sites. For the composite of PCL-HA, hydroxyapatite was observed to be adsorbed and form crystals in the form of needles on the inner surface of the membrane. For the composite of PCL-α-Al_2_O_3_, mineralization and certain nodules that have different morphologies produced by the cells can be observed; alpha alumina allows the adhesion, proliferation, and mineralization process. The production of crystals or nodules is lower because alpha alumina does not present chemical interaction directly with the cells. PCL-HA-α-Al_2_O_3_ favors cell adhesion and proliferation, and allows mineralization, forming larger crystals in contrast to those present in the PCL-α-Al_2_O_3_ composite. A collagen fiber can be observed, entangled in the fibers of the composites according to histological studies for the start and development of mineral formation [26].

## 4. Conclusions

The PCL, PCL-HA, PCL-α-Al_2_O_3_, and PCL-HA-α-Al_2_O_3_ nanofibers were manufactured by electrospinning technique with an approximate size of (0.84 ± 0.23) μm. The diameter of the nanofibers increased with the addition of the same ceramic particles that caused a roughness in the surface of the nanofibers. The characterization of the nanofibers and nanopowders of ceramic particles showed that the ceramic particles are embedded in the PCL. Additional bands for chemical bonds cannot be observed in infrared spectroscopy, showing a physical union. The subcutaneous tissues of Wistar rats were evaluated at 2, 4, and 6 weeks, observing a chronic inflammatory infiltrate—a characteristic foreign body reaction—which decreased slightly at 6 weeks with the addition of ceramic particles. It is said that the manufactured composites are not totally biocompatible since a foreign body reaction was presented; however, as tissue loss and necrosis were not observed and there was a slight decrease in the inflammatory infiltrate in the last evaluation period, it can be used for the beginning of an acceptance of the organism for the tested materials. This preliminary study shows adjustments for future research, such as the application of the material on a membrane and not on a roll, which would help the contact of the material with the tissue of Wistar rats.

## Figures and Tables

**Figure 1 polymers-14-02130-f001:**
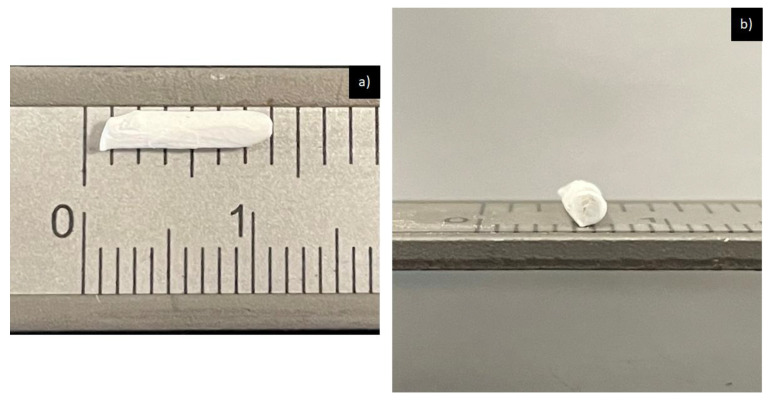
PCL implant before being implanted in the animal: (**a**) implant observed along; (**b**) implant observed from its internal diameter.

**Figure 2 polymers-14-02130-f002:**
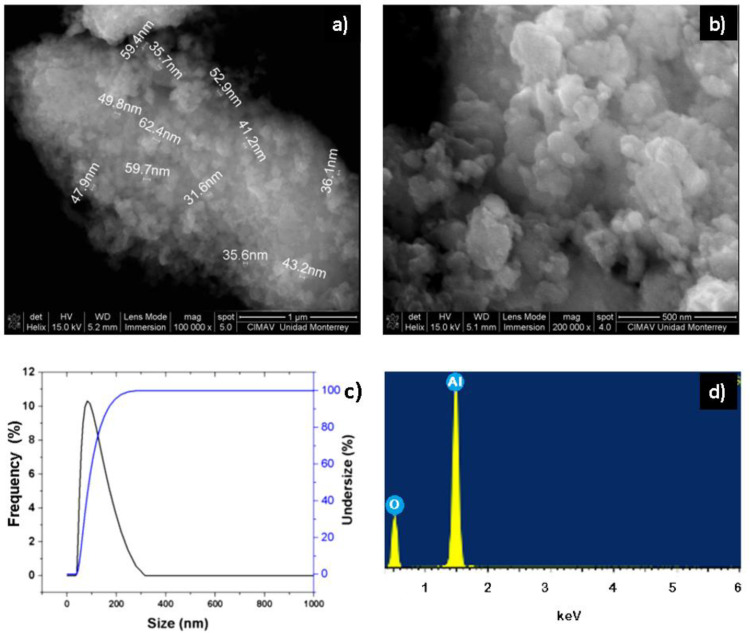
SEM micrographs at (**a**) 100,000× and (**b**) 200,000×; (**c**) DLS and (**d**) EDX spectrum for α-Al_2_O_3_ nanopowders.

**Figure 3 polymers-14-02130-f003:**
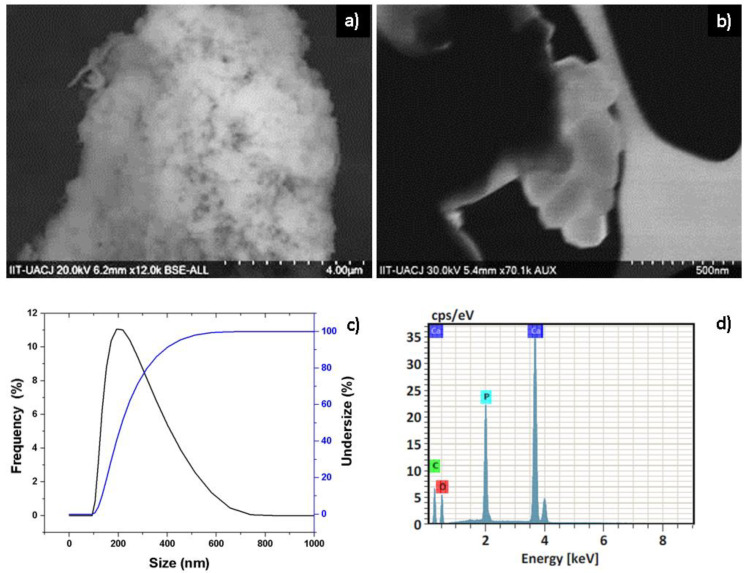
SEM micrographs at (**a**) 12,000× and (**b**) 70,000×; (**c**) DLS and (**d**) EDX spectrum for hydroxyapatite nanopowders.

**Figure 4 polymers-14-02130-f004:**
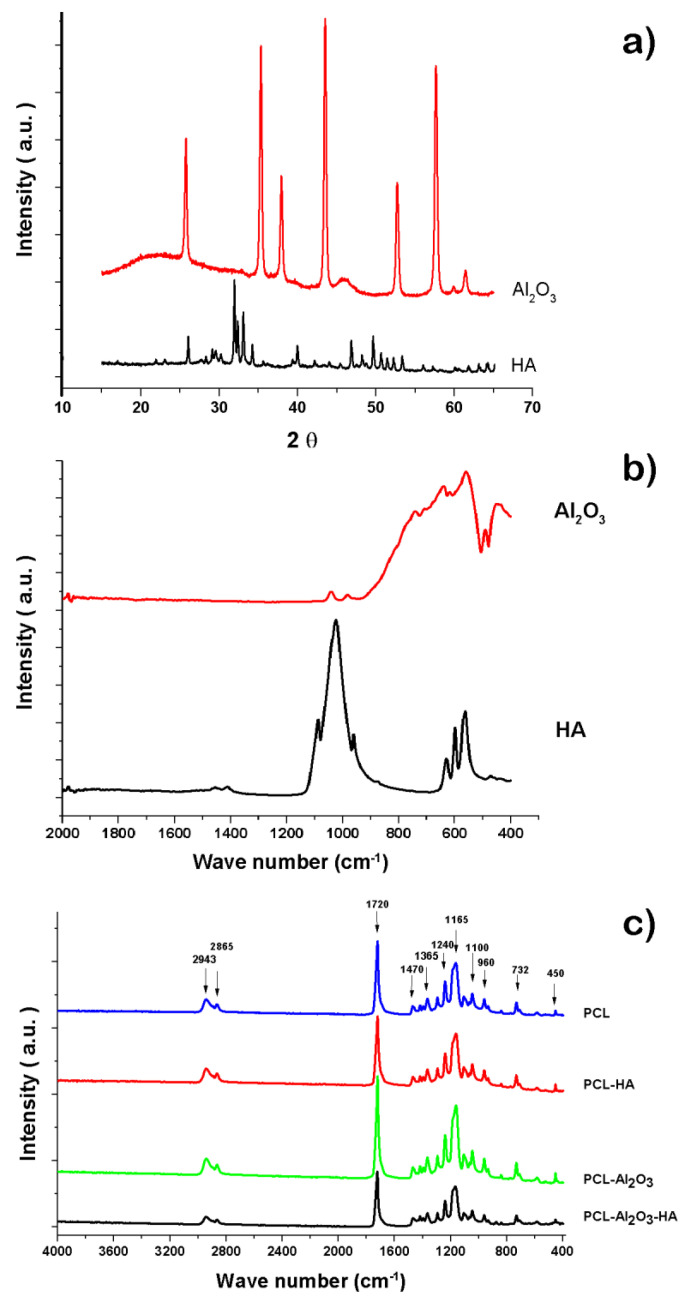
(**a**,**b**) Infrared spectrum obtained from the nanopowders of hydroxyapatite and α-Al_2_O_3_; (**c**) infrared spectrum obtained from the different nanofibers PCL, PCL-HA, PCL-α-Al_2_O_3_, and PCL-HA-α-Al_2_O_3_.

**Figure 5 polymers-14-02130-f005:**
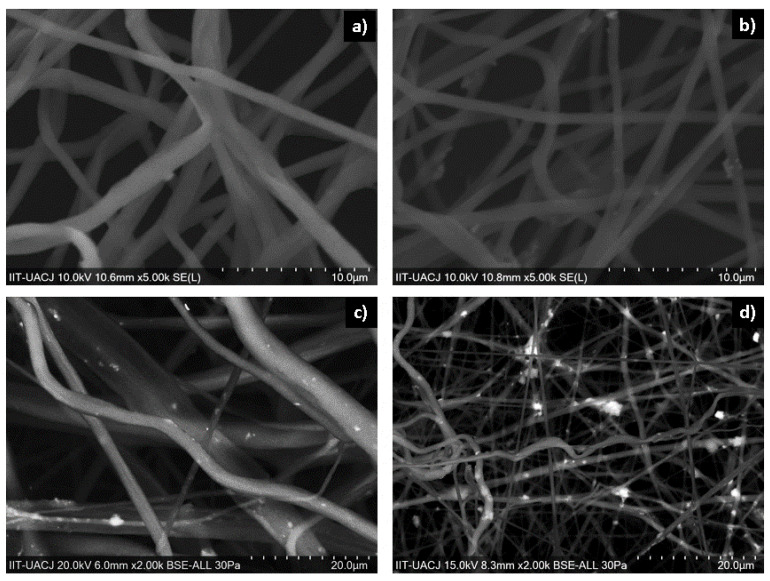
Micrographs obtained by SEM: (**a**) PCL, (**b**) PCL-HA, (**c**) PCL-α-Al_2_O_3_, and (**d**) PCL-HA-α-Al_2_O_3_ fibers.

**Figure 6 polymers-14-02130-f006:**
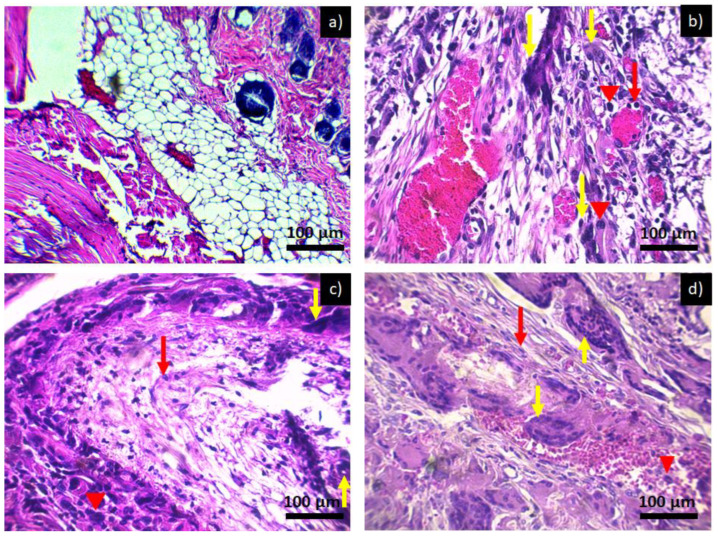
Histological preparation in tissue hematoxylin-eosin with the PCL: (**a**) control, (**b**) PCL 2 weeks, (**c**) PCL 4 weeks, and (**d**) PCL 6 weeks. Red arrows indicate the presence of lymphocytes, red arrowheads indicate macrophages, and yellow arrows indicate the presence of multinucleated giant cells characteristic of a chronic inflammatory infiltrate.

**Figure 7 polymers-14-02130-f007:**
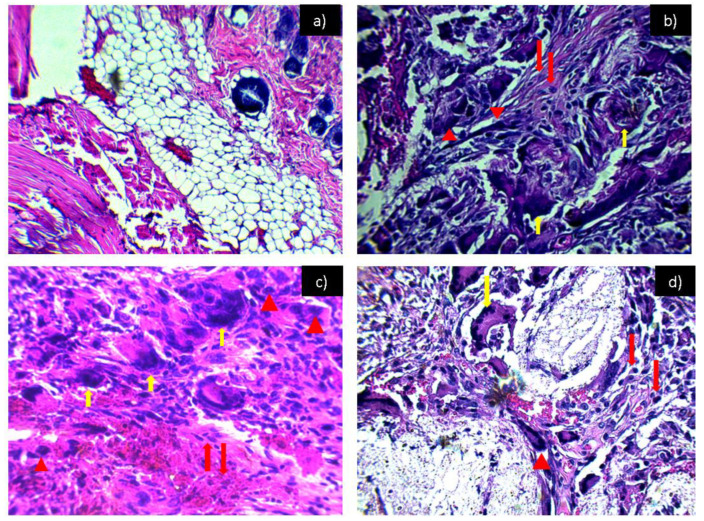
Histological preparation in tissue hematoxylin-eosin with the PCL/HA: (**a**) control, (**b**) PCL/HA 2 weeks, (**c**) PCL/HA 4 weeks, and (**d**) PCL/HA 6 weeks. Red arrows indicate the presence of lymphocytes, red arrowheads indicate macrophages, and yellow arrows indicate the presence of multinucleated giant cells characteristic of a chronic inflammatory infiltrate.

**Figure 8 polymers-14-02130-f008:**
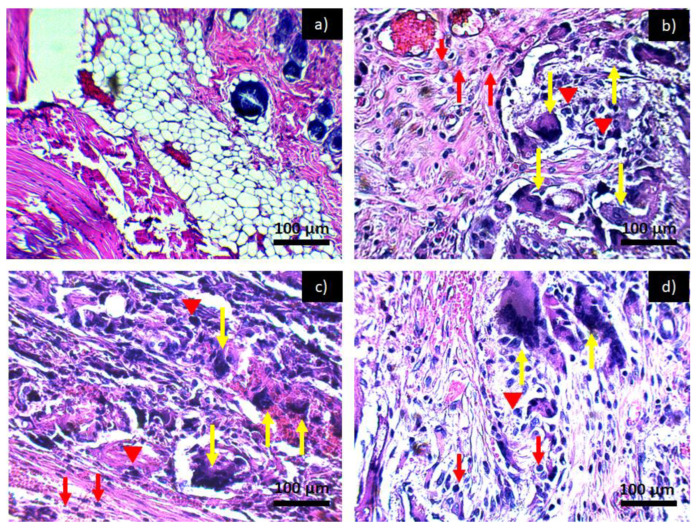
Histological preparation in tissue hematoxylin-eosin with the PCL/α-Al_2_O_3_: (**a**) control, (**b**) PCL/α-Al_2_O_3_ 2 weeks, (**c**) PCL/α-Al_2_O_3_ 4 weeks, and (**d**) PCL/α-Al_2_O_3_ 6 weeks. Red arrows indicate the presence of lymphocytes, red arrowheads indicate macrophages, and yellow arrows indicate the presence of multinucleated giant cells characteristic of a chronic inflammatory infiltrate.

**Figure 9 polymers-14-02130-f009:**
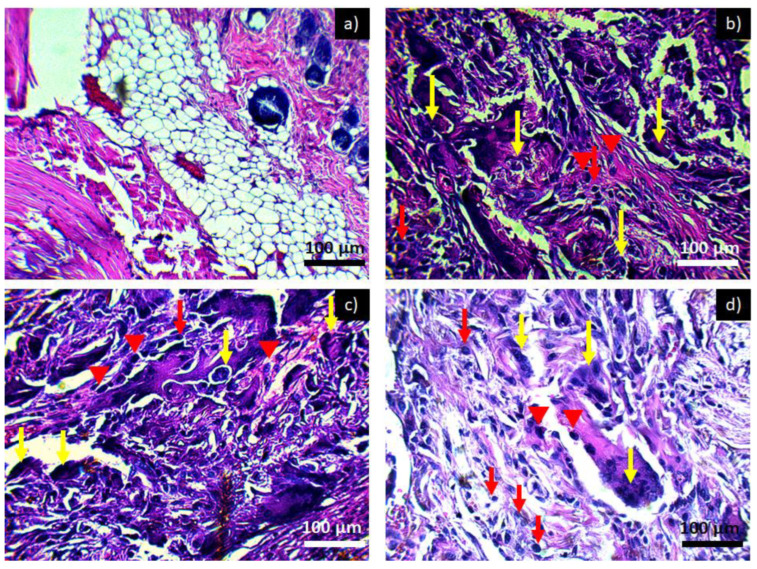
Histological preparation in tissue hematoxylin-eosin with the PCL/HA-α-Al_2_O_3_: (**a**) control, (**b**) PCL/HA-α-Al_2_O_3_ 2 weeks, (**c**) PCL/HA-α-Al_2_O_3_ 4 weeks, and (**d**) PCL/HA-α-Al_2_O_3_ 6 weeks. Red arrows indicate the presence of lymphocytes, red arrowheads indicate macrophages, and yellow arrows indicate the presence of multinucleated giant cells characteristic of a chronic inflammatory infiltrate.

**Figure 10 polymers-14-02130-f010:**
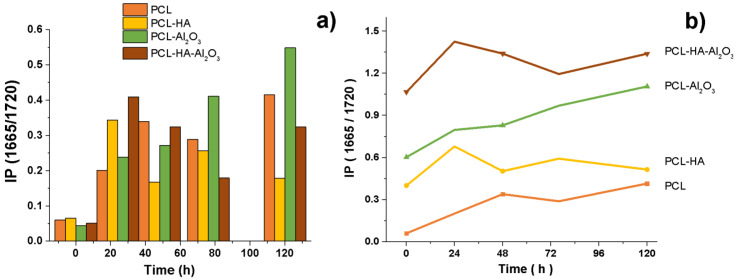
IP(1660/1720) for the IR spectra of cellular proliferation on PCL, PCL-HA, PCL- α-Al_2_O_3_, and PCL-HA-α-Al_2_O_3_ (coulomb form (**a**) and stacked lines (**b**)).

**Figure 11 polymers-14-02130-f011:**
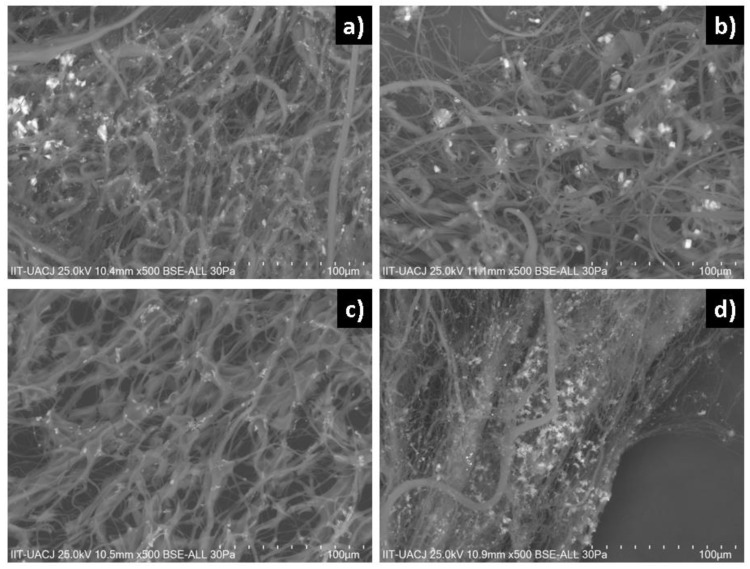
Morphology of mineralization of the fibers: (**a**) PCL, (**b**) PCL-HA, (**c**) PCL-α-Al_2_O_3_, and (**d**) PCL-HA-α-Al_2_O_3_.

**Table 1 polymers-14-02130-t001:** Percentage of Cell viability test data.

	Composite			
Time (h)	PCL	PCL-HA	PCL-α-Al_2_O_3_	PCL-HA-α-Al_2_O_3_
24	14 ± 1.7%	14 ± 3.1%	43 ± 2.3%	52 ± 3.6%
48	22 ± 2.4%	89 ± 4.8%	39 ± 3.6%	61 ± 3.1%
76	127 ± 5.1%	131 ± 4.4%	119 ± 5.8%	138 ± 4.4%

## Data Availability

The data presented in this study are available on request from the corresponding author.

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
