# Peer review of "Poly-ε-Caprolactone-Hydroxyapatite-Alumina (PCL-HA-α-Al2O3) Electrospun Nanofibers in Wistar Rats"

_polymers, 2022, doi:10.3390/polym14112130_

Round 1

Reviewer 1 Report

Dear Authors,

in your interesting manuscript, the following points should be added/changed to further improve it:

  • line 26: averages +- standard deviations must be given with brackets or with doubled units, e.g. (840 +- 230) nm, else the average wouldn't have a unit (ditto in the residual text).
  • Comparing the mail text with the headline, the "alpha" before "alumina" is missing.
  • line 145: Do you mean "1000 W"?
  • line 161: The "y" is an "and" ;-)
  • line 223: Which eosin, yellow or red?
  • The first half sentence from section 3 belongs into section 2. Please add more information about the instruments (names, producers) you used. And please define the abbreviations.
  • Line 238: Please define STEM.
  • Many figures are not sharp, e.g. Fig. 2c and d. Besides, what about gold or whatever you used for sputtering in Fig. 2d, and what is "undersize" in Fig. 2c?
  • Fig. 4: Both "intensity" have different sizes. The wavenumbers inside Fig. 4b are too small to be readable.
  • Discussion of Fig. 6: How does the process look like in the reference rat?
  • line 370 ff: Please add the standard deviations, else it is not possible to evaluate whether these values differ for different materials or not. Why is the viability so low after incubation for one day?
  • Fig. 10: How many rats were used? If it is more than 1 per test, please add standard deviations.
  • line 479: The acknowledgment should either be filled or deleted.

Author Response

Thank you very much for the observations the corrections were already made in the manuscript

line 26: averages +- standard deviations must be given with brackets or with doubled units, e.g. (840 +- 230) nm, else the average wouldn't have a unit (ditto in the residual text).

Response: parentheses are placed around the means and standard deviations

Comparing the mail text with the headline, the "alpha" before "alumina" is missing.

Response: “Alpha” was placed before “alumina”

 line 145: Do you mean "1000 W"?

Response: The manufacturer indicates that the oven is 1000W W=Watt

  line 161: The "y" is an "and" ;-)

Response: It was corrected

  line 223: Which eosin, yellow or red?

Response: red eosin was used

 The first half sentence from section 3 belongs into section 2. Please add more information about the instruments (names, producers) you used. And please define the abbreviations.

Response: it was corrected in the text section 2

 Line 238: Please define STEM.

Response: it was corrected in the text

Many figures are not sharp, e.g. Fig. 2c and d. Besides, what about gold or whatever you used for sputtering in Fig. 2d, and what is "undersize" in Fig. 2c?

Response: Images were corrected, undersize is the accumulated percentage of particles of different sizes

  Fig. 4: Both "intensity" have different sizes. The wavenumbers inside Fig. 4b are too small to be readable.

Response: Images were corrected,

 Discussion of Fig. 6: How does the process look like in the reference rat?

Response: In each of the images, the reference tissue is shown, which does not show the presence of inflammatory cells and is compared with the tissues exposed to the materials.

 line 370 ff: Please add the standard deviations, else it is not possible to evaluate whether these values differ for different materials or not. Why is the viability so low after incubation for one day?

Response: The information was corrected, and next paragraph is added in the text

 Fig. 10: How many rats were used? If it is more than 1 per test, please add standard deviations.

Response: Only used one rat

line 479: The acknowledgment should either be filled or deleted.

Response: The section was corrected

Reviewer 2 Report

  1. The qualities of figures are very coarse, especially in Figure 2 and Figure 3.
  2. Line 151 “The solutions were prepared from the dilution of poly-ε caprolactone (PCL) (ACS, 150 Sigma-Aldrich ©, 80,000 kDa) in a 10% acetone”

What is the concentration of PCL? What is 10% acetone?

  1. There are many typos in the manuscript. Ex: Line 161 “PCL-α-Al2O3 y PCL-HA-α-Al2O3.”……….
  2. Line 118 “Three fractions were taken from each concentration of the nanofibers. The implants measured approximately 8-10 mm in length by 1.3 mm in internal diameter.

What is the “each concentration of the nanofibers”

How to fabricate the cylinder implant?

  1. Line 238 “The characterization of hydroxyapatite nanopowders by SEM and STEM as shown in Figure 3a and 3b”

Figure 3b is SEM not STEM.

  1. Figure 258 “Figure 4a (black) shows the infrared spectrum for the α-Al2O3 nanopowders for bands corresponding to the flexural stretching of the Al-O bond at 390 cm-1"

However, the wavelength number is from 4000 to 400 cm-1 in Figure 4a

  1. It is necessary to demonstrate the phase of hydroxyapatite powder, Al2O3 powder and the different nanofibers using XRD.
  2. Line 254~261. Please mark the characteristic peaks in Figure 4a.
  3. The diameter of the PCL-α-Al2O3 and PCL-HA-α-Al2O3 are 1.13 ± 0.41 um and 1.39 ± 0.64 um, respectively. According to the Figure 5a and Figure 5b, the diameter of PCL-HA-α-Al2O3 nanofibers is smaller than the PCL-α-Al2O3 nanofibers.
  4. Figure 4a and Figure 10 are same with the reference 17.

Author Response

Thank you very much for the observations the corrections were already made in the manuscript

Revisor 2

  1. The qualities of figures are very coarse, especially in Figure 2 and Figure 3.

Response: Images were corrected,

  1. Line 151 “The solutions were prepared from the dilution of poly-ε caprolactone (PCL) (ACS, 150 Sigma-Aldrich ©, 80,000 kDa) in a 10% acetone”

What is the concentration of PCL? What is 10% acetone?

Response: the concentration of pcl dissolved in the acetone at 10% w/v

There are many typos in the manuscript. Ex: Line 161 “PCL-α-Al2O3 y PCL-HA-α-Al2O3.”……….

  1. Line 118 “Three fractions were taken from each concentration of the nanofibers. The implants measured approximately 8-10 mm in length by 1.3 mm in internal diameter.

What is the “each concentration of the nanofibers”

Response: refers to each of the electrospun materials, has been corrected in the text

The concentration of powders is 2 %.(line 155)

How to fabricate the cylinder implant?

Response: The cylindrical implant was manufactured by rolling the membrane obtained from electrospinning.

  1. Line 238 “The characterization of hydroxyapatite nanopowders by SEM and STEM as shown in Figure 3a and 3b”

Figure 3b is SEM not STEM.

Response: Figures 3a and 3b are STEM images at 12,000 and 70000 X

  1. Figure 258 “Figure 4a (black) shows the infrared spectrum for the α-Al2O3 nanopowders for bands corresponding to the flexural stretching of the Al-O bond at 390 cm-1"

However, the wavelength number is from 4000 to 400 cm-1 in Figure 4a

Response: The wave number is 440 cm-1 has been corrected in the text

It is necessary to demonstrate the phase of hydroxyapatite powder, Al2O3 powder and the different nanofibers using XRD.

Response: The characterization is previously reported and cited. 

  1. Line 254~261. Please mark the characteristic peaks in Figure 4a.
  2. The diameter of the PCL-α-Al2O3 and PCL-HA-α-Al2O3 are 1.13 ± 0.41 um and 1.39 ± 0.64 um, respectively. According to the Figure 5a and Figure 5b, the diameter of PCL-HA-α-Al2O3 nanofibers is smaller than the PCL-α-Al2O3 nanofibers.

Response: The characterization of diameters is from previously reported and cited. 

The measurements of the diameters of the fibers were taken from 500 fibers at random in different fields, therefore the size with its standard deviation is reported.

  1. Figure 4a and Figure 10 are same with the reference 17.

Response: Figure 4a is replaced with new.

Round 2

Reviewer 1 Report

Dear Authors,

after the revisions, the following points remain open:

Line 166: The JEOL JSM-6400 is actually an SEM. Besides, JEOL produces also TEMs. There is no such such like a "scanning transmission electron microscope", it either "scanning" or "transmission". If you have a special version of the JEOL JSM-6400 which somehow includes both SEM and TEM, please indicate this. However, since the bottom line of the images in Figs. 2 and 3 is different, I assume you used a second instrument (and should thus also be corrected in the caption of Fig. 3).

Many figures are still not sharp. Please insert the original images in full size. Besides, what about gold or whatever you used for sputtering in Fig. 2d?

> Response: Images were corrected, undersize is the accumulated percentage of particles of different sizes

This doesn't make sense; undersize normally means something else.

Fig. 4: The wavenumbers inside Fig. 4b are too small to be readable.

> line 370 ff: Please add the standard deviations, else it is not possible to evaluate whether these values differ for different materials or not. Why is the viability so low after incubation for one day?

> Response: The information was corrected, and next paragraph is added in the text

I see some information about any previous ANOVA, but not the answer to my question. It is complicated enough if values are only given in the text and not in a graph or table, so what we actually need is a graph or table with the values you discuss incl. standard deviations: viability after 24 h, 48 h, and 72 ours of incubation for reference, PCL and PCL-HA. Regarding the newly added information: How did you test that an ANOVA can be used here, what do you mean with this significance, what is the unit of the (standard?) deviation, what is F, and where do all these degrees of freedom stem from?

Author Response

Revisor 1

Line 166: The JEOL JSM-6400 is actually an SEM. Besides, JEOL produces also TEMs. There is no such such like a "scanning transmission electron microscope", it either "scanning" or "transmission". If you have a special version of the JEOL JSM-6400 which somehow includes both SEM and TEM, please indicate this. However, since the bottom line of the images in Figs. 2 and 3 is different, I assume you used a second instrument (and should thus also be corrected in the caption of Fig. 3).

Response: text is corrected, Its SEM whit Scanning transmission electron microscopy detector

Many figures are still not sharp. Please insert the original images in full size. Besides, what about gold or whatever you used for sputtering in Fig. 2d?

>Response: Images were corrected, undersize is the accumulated percentage of particles of different sizes

This doesn't make sense; undersize normally means something else.

Response: The DLS technique provides an average picture of the sample by determining the size of millions of particles.

Fig. 4: The wavenumbers inside Fig. 4b are too small to be readable.

Response: the image is corrected,

> line 370 ff: Please add the standard deviations, else it is not possible to evaluate whether these values differ for different materials or not. Why is the viability so low after incubation for one day?

> Response: The information was corrected, the standard deviations is added (line 373 to 378)

> Response: The information was corrected, and next paragraph is added in the text

I see some information about any previous ANOVA, but not the answer to my question. It is complicated enough if values are only given in the text and not in a graph or table, so what we actually need is a graph or table with the values you discuss incl. standard deviations: viability after 24 h, 48 h, and 72 ours of incubation for reference, PCL and PCL-HA. Regarding the newly added information: How did you test that an ANOVA can be used here, what do you mean with this significance, what is the unit of the (standard?) deviation, what is F, and where do all these degrees of freedom stem from?

Reviewer 2 Report

  1. The qualities of images still were not improved in Figure 2 and Figure 3.
  2. The characterization of hydroxyapatite powder and the different nanofibers is previously reported and cited at reference 17. However, it is lack of XRD data to demonstrate the phase components.
  3. The only one rat was used in the biocompatibility test in vivo.

Is this feasible in in-vivo test?

  1. The diameter of the PCL-α-Al2O3and PCL-HA-α-Al2O3 are 1.13 ± 0.41 um and 1.39 ± 0.64 um, respectively. According to the Figure 5a and Figure 5b, the diameter of PCL-HA-α-Al2O3 nanofibers is smaller than the PCL-α-Al2O3 nanofibers

Response: The characterization of diameters is from previously reported and cited.  The measurements of the diameters of the fibers were taken from 500 fibers at random in different fields, therefore the size with its standard deviation is reported.

It is necessary to show the low magnification of SEM images in Figure 5

  1. Figure 10 is still same with the figure 8e of reference 17.

Author Response

Revisor 2

  1. The qualities of images still were not improved in Figure 2 and Figure 3.

Response: the images is corrected,

  1. The characterization of hydroxyapatite powder and the different nanofibers is previously reported and cited at reference 17. However, it is lack of XRD data to demonstrate the phase components.

Response: the XRD data information is added,

  1. The only one rat was used in the biocompatibility test in vivo.Is this feasible in in-vivo test?

Response: The studies were done in triplicate, but a single rat was given a sample of each material.

  1. The diameter of the PCL-α-Al2O3and PCL-HA-α-Al2O3 are 1.13 ± 0.41 um and 1.39 ± 0.64 um, respectively. According to the Figure 5a and Figure 5b, the diameter of PCL-HA-α-Al2O3 nanofibers is smaller than the PCL-α-Al2O3 nanofibers

Response: The characterization of diameters is from previously reported and cited.  The measurements of the diameters of the fibers were taken from 500 fibers at random in different fields, therefore the size with its standard deviation is reported.

It is necessary to show the low magnification of SEM images in Figure 5

Response: the images shown are the most representative where the dimensions of the fibers are better seen, at lower magnifications only the membrane would be seen, at higher magnifications we would only see a few fibers, in addition the measurements of the diameters of the fibers were in several fields, it is what recommended. most studies report measurements of 100 to 200 fibers, an analysis with 500 measurements is more accurate and gives less deviation

Figure 10 is still same with the figure 8e of reference 17.

Response: the images is corrected,

Round 3

Reviewer 1 Report

Dear Authors,

the following points remain identical:

  • Most graphs and other images are stil not sharp. Besides, I just recognized (by zooming in, else it is not fully readable) that you misspelled "frequency" on the y-axes.
  • In lines 376 ff, the values are still confusing. As mentioned last time - please prepare a proper table. I don't see the viability values for the control cells, so I cannot tell wether they are really increased for PCL etc. Besides, please give standard deviations and averages correctly - here again the brackets are missing, and both must have the same accuracy.
  • The ANOVA part ist still not explained at all. And actually it is not necessary since it contains no information.

Author Response

Most graphs and other images are stil not sharp. Besides, I just recognized (by zooming in, else it is not fully readable) that you misspelled "frequency" on the y-axes.

Response : I can send original files if is necessary.

In lines 376 ff, the values are still confusing. As mentioned last time - please prepare a proper table. I don't see the viability values for the control cells, so I cannot tell wether they are really increased for PCL etc. Besides, please give standard deviations and averages correctly - here again the brackets are missing, and both must have the same accuracy.

The ANOVA part ist still not explained at all. And actually it is not necessary since it contains no information.

Response: A new table is added

Reviewer 2 Report

No comment.

Author Response

I check